# Compiler Auto-Vectorization with Imitation Learning

**Charith Mendis**
MIT CSAIL
charithm@mit.edu

**Cambridge Yang**
MIT CSAIL
camyang@csail.mit.edu

**Yewen Pu**
MIT CSAIL
yewenpu@mit.edu

**Saman Amarasinghe**
MIT CSAIL
saman@csail.mit.edu

**Michael Carbin**
MIT CSAIL
mcarbin@csail.mit.edu

## Abstract

Modern microprocessors are equipped with single instruction multiple data (SIMD) or vector instruction sets which allow compilers to exploit fine-grained data level parallelism. To exploit this parallelism, compilers employ *auto-vectorization* techniques to automatically convert scalar code into vector code. Larsen & Amarasinghe (2000) first introduced superword level parallelism (SLP) based vectorization, which is a form of vectorization popularly used by compilers. Current compilers employ hand-crafted heuristics and typically only follow one SLP vectorization strategy which can be suboptimal. Recently, Mendis & Amarasinghe (2018) formulated the instruction packing problem of SLP vectorization by leveraging an integer linear programming (ILP) solver, achieving superior runtime performance. In this work, we explore whether it is feasible to imitate optimal decisions made by their ILP solution by fitting a graph neural network policy. We show that the learnt policy, *Vemal*, produces a vectorization scheme that is better than the well-tuned heuristics used by the LLVM compiler. More specifically, the learnt agent produces a vectorization strategy that has a 22.6% higher average reduction in cost compared to the LLVM compiler when measured using its own cost model, and matches the runtime performance of the ILP based solution in 5 out of 7 applications in the NAS benchmark suite.

## 1  Introduction

Modern microprocessors have introduced single instruction multiple data (SIMD) or vector units (e.g. Intel x86 AVX extensions[1]) to accelerate execution of performance critical applications by performing computations on multiple data items in parallel. In order to use these vector units, programmers must either code using platform specific vector assembly instructions (Figure 1(c)), which is tedious, error-prone and results in non-portable code or use existing compiler *auto-vectorization* techniques to automatically discover data parallel portions of programs and to transform scalar instructions (Figure 1(a)) in such regions into vector instructions (Figure 1(b),(c)).

Larsen & Amarasinghe (2000) first showed how to perform compiler auto-vectorization using fine grained parallelism available in programs at the instruction level (superword level parallelism) targeting fixed-width vector instructions available in modern microprocessors. Superword level parallelism (SLP) based vectorization provided a more general alternative to loop vectorization techniques proposed earlier (Allen & Kennedy, 1987; Sreraman & Govindarajan, 2000).

In order to exploit SLP for vectorization, compilers need to select a set of scalar instructions which can be executed in parallel and merge them into a single vector instruction. It is shown that this process reduces to the optimal subset selection problem, which is known to be NP-hard. Therefore, many existing SLP vectorization schemes are driven by hand-crafted heuristics (Liu et al., 2012;

```
a[0] = b[0] + c[0]                                        movdqa  xmm0, c
                          {a[0],a[1]} = {b[0],b[1]} + {c[0],c[1]}    paddq   xmm0, b
a[1] = b[1] + c[1]                                        movdqa  a, xmm0
        (a)                         (b)                              (c)
```

Figure 1: (a) scalar code storing result of adding elements loaded from array `b`,`c` into array `a` (b) compiler auto-vectorized code; expressions inside {.} are executed in parallel. (c) manually written vector assembly instructions / executable code generated by the compiler for auto-vectorized code (b)

Shin et al., 2003, 2005, 2002), which can be suboptimal. Recently, goSLP (Mendis & Amarasinghe, 2018) introduced a SLP vectorization strategy with certain optimality guarantees guided by an Integer Linear Programming (ILP) solver. It has shown superior vectorization schemes compared to heuristics guided solutions, achieving end-to-end speedups on well-known compiler benchmark suites.

In this work, we propose a technique to learn how to vectorize by imitating the solution presented in goSLP (Mendis & Amarasinghe, 2018). Specifically, we formulate the decision procedure as a Markov Decision Process (MDP) and then use the DAGGER algorithm (Ross et al., 2011) to collect traces on how the ILP solver solves the SLP vectorization problem. We use these trace aggregates as supervision to train a parametrized policy modeled by a Gated Graph Neural Network (Li et al., 2015; Allamanis et al., 2017).

We show that the learnt policy, *Vemal*, outperforms one of the well-tuned heuristics used in the LLVM compiler (Lattner & Adve, 2004) both in terms of static metrics and dynamic runtime performance, while matching the performance of goSLP in 5 out of 7 programs in the NAS benchmark suite held out for testing.

Specifically, we make the following contributions:

- Formulation of the SLP vectorization problem as a Markov Decision Process (MDP), where the vectorization strategy is constructed sequentially.
- Modeling of the agent policy that solves the MDP as a Gated Graph Neural Network (GGNN) (Li et al., 2015; Allamanis et al., 2017) and using imitation learning to train the policy network to mimic the optimal decisions made by the ILP solver.
- Evaluation of the learnt policy on representative compiler benchmark suites (SPEC2006fp C/C++ (Henning, 2006), SPEC2017fp C/C++ (Bucek et al., 2018) and NAS benchmark suites (Bailey et al., 1991)). Specifically, we show that the learnt policy has an average static cost reduction which is 22.6% higher than LLVM as measured by LLVM's own static cost model. We also show that the learnt policy achieves a geometric mean runtime speedup of $1.015\times$ on the NAS benchmark suite (held out during training) compared to LLVM, matching runtime speedups of goSLP in 5 out of 7 applications.

In summary, we have shown that it is possible to learn end-to-end compiler optimization policies which surpass the performance of hand-crafted compiler heuristics by imitating an optimal solution.

## 2  The SLP Vectorization Problem

Superword Level Parallelism (SLP) is a type of fine-grained parallelism first introduced by Larsen & Amarasinghe (2000) that is suitable for vector code generation. SLP is available in cases where two or more scalar instructions are *independent* (the second instruction does not require the output of the first instruction, and vice-versa) and are *isomorphic* (of the same instruction type, such as addition or subtraction). For example, consider the code snippet from Figure 2(a). It computes intermediate values $A_1$ up to $A_3$ by dividing values loaded from array $L$. Next, these intermediate values are used in a series of subtraction operations and the results are stored in array $S$, which is disjoint from $L$. Both the division and subtraction instruction groups, adjacent loads and stores are independent and isomorphic. Hence, they are amenable to SLP vectorization.

Such scalar instructions can be executed in parallel and hence can be *packed* to form a single vector instruction. We name a set of scalar instructions which are packed as a *vector pack*. For example considering the code snippet shown in Figure 2(a), we can form a vector pack out of $A_1$ and $A_2$ and perform the division in vector form as: $\{A_1, A_2\} = \{L[5], L[6]\} / \{L[2], L[3]\}$.

Forming vector instructions by packing multiple scalar instructions can lead to better runtime performance. However, not all packing opportunities are profitable. For instance, if a set of instructions is packed while their operands remain scalar, there will be an additional packing overhead to bring the

operands also into packed form. On the other hand, if the output of packed instructions is used by other scalar instructions, there will be an unpacking overhead.

The task of the compiler is to find the most profitable set of vector packs to form among all available packing opportunities. This combinatorial optimization task is not easy even in the case of pairwise instruction packing (Mendis & Amarasinghe, 2018) and is NP-hard.

## 2.1 Motivating Example

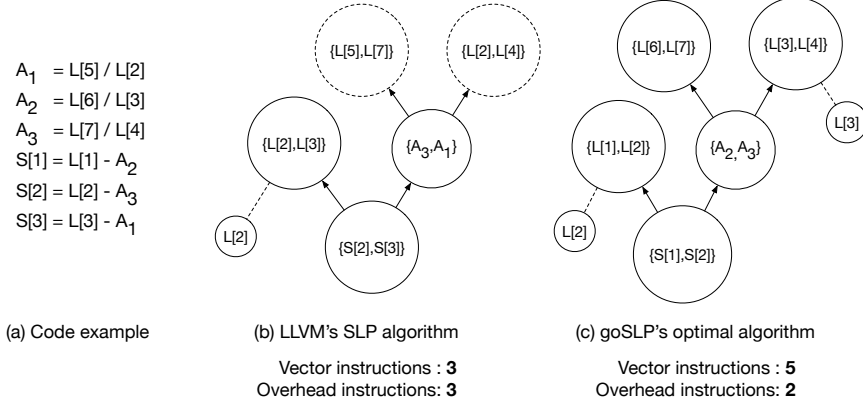

A_1 = L[5] / L[2]
A_2 = L[6] / L[3]
A_3 = L[7] / L[4]
S[1] = L[1] - A_2
S[2] = L[2] - A_3
S[3] = L[3] - A_1

(a) Code example    (b) LLVM's SLP algorithm    (c) goSLP's optimal algorithm

Vector instructions : **3**          Vector instructions : **5**
Overhead instructions: **3**        Overhead instructions: **2**

Figure 2: Comparison of SLP vectorization strategies (a) code example; $S$ and $L$ are two disjoint arrays, (b)-(c) show dependency graphs of packed instructions under each instruction packing strategy (b) under LLVM's SLP vectorization algorithm (c) optimal packing. Solid arrows show dependencies. Groupings with solid circles show vector packs. Groupings with dotted circles show packs created using overhead packing instructions and dotted lines show unpacking of values from vector packs.

We compare LLVM's greedy packing strategy with goSLP's optimal packing strategy for the code snippet shown in Figure 2(a) to illustrate how instruction packing affects the quality of vectorization.

Figure 2(b) shows the packing solution of LLVM's SLP vectorization algorithm. LLVM's greedy algorithm chooses to vectorize adjacent stores $S[2]$ and $S[3]$ first. Next, it tries to pack the operands of already packed values recursively. Using this procedure, operands of $\{S[2], S[3]\}$ are packed next, forming vector packs $\{L[2], L[3]\}$ and $\{A_1, A_3\}$. However, operands of $\{A_1, A_3\}$ cannot be packed, since they access non-adjacent memory locations, forcing LLVM to emit two overhead packing instructions. Further, $L[2]$ needs to be unpacked before it can be repacked with $L[4]$. Altogether, LLVM's strategy yields 3 vector instructions while incurring a cost of 3 overhead instructions.

Figure 2(c) shows the optimal packing solution found by goSLP using an ILP solver. The optimal strategy creates vector pack $\{S[1], S[2]\}$ instead of $\{S[2], S[3]\}$, which proves to be globally optimal. Altogether, this strategy yields 5 vector instructions and requires only 2 overhead instructions.

In this paper, we evaluate whether it is possible to learn a policy that imitates goSLP's ILP solution for the pairwise instruction packing problem. Note that, we are not aiming to learn a general purpose ILP solver, but to imitate the ILP solution for this problem domain. We will now formalize the pairwise instruction packing problem.

## 2.2 The Pairwise Instruction Packing Problem

Let $I = \{I_1, I_2, ..., I_{n-1}, I_n, I_\epsilon\}$ be the set of $n + 1$ instructions, where $I_\epsilon$ is an artificial empty Instruction. A valid instruction packing $\bar{P} = \{P_1 \dots P_m\}$ is a collection of pairs:

$$P_i \in I \times I \ , \ \bigcup_i P_i = I$$

that satisfies two kinds of packing constraints: $C_I$ (within pairs) and $C_{II}$ (between pairs).

**Within Pair** A legal pack $P = (I_i, I_j)$ must satisfy the within pair constraint $C_I(P)$

- $I_i$ and $I_j$ must be isomorphic: perform the same operation on same data types which results in values of the same type.
- $I_i$ and $I_j$ must be independent: $I_i$ and $I_j$ cannot be directly or transitively dependent, where they cannot be reachable by one another in the same data-dependency graph.

- If $I_i$ and $I_j$ require reordering, it should be possible under the hardware memory model.
- If $I_i$ and $I_j$ access memory they must access adjacent memory locations.
- $I_i \prec I_j$ under some ordering of statements $I_1 \prec \cdots \prec I_n \prec I_\epsilon$

**Between Pairs** packs $P_i$ and $P_j$ can both be created iff they satisfy the between pairs constraint $C_{\mathrm{II}}(P_i, P_j)$:

- $P_i$ and $P_j$ are schedulable: there shouldn't be any circular dependencies between the two packs. For example, if $I_{i,1}, I_{i,2} \in P_i$ and $I_{j,1}, I_{j,2} \in P_j$, it shouldn't be the case that $I_{i,1} \, \delta \, I_{j,1}$ and $I_{j,2} \, \delta \, I_{i,2}$, where $\delta$ denotes dependency.
- $P_i$ and $P_j$ are not overlapping: $\forall I_i \in P_i \implies I_i \notin P_j$. That is, a single statement can only belong to one pack.

We write $C_{\mathrm{II}}(P_i, P_j)$, if the packs satisfy the between pairs constraint.

**Static Cost.** We evaluate the efficacy of the packing strategies using LLVM's static cost model. The static cost model assigns costs for each vector and scalar instruction. The total cost of a block of code is calculated as the addition of these per instruction costs.

**Objective.** Let $F$ be the performance (measured using static cost) of executing the block of code with vectorized instructions formed by a packing. The goal of any SLP vectorization scheme is to find a packing strategy $P$ that minimizes the cost ($\operatorname{argmin}_{\bar{P}} F(\bar{P})$) subject to constraints $C_{\mathrm{I}}$ and $C_{\mathrm{II}}$.

## 3 Learnt Solution - Vemal

Our learnt vectorization policy, Vemal, imitates goSLP's (Mendis & Amarasinghe, 2018) ILP solution to perform SLP vectorization. We cast the pairwise instruction packing problem as a Markov Decision Process (MDP), which we solve by a neural-network policy that imitates the ILP solution via imitation learning using the DAGGER algorithm (Ross et al., 2011).

### 3.1 MDP formulation

We formulate the pairwise instruction packing problem as a MDP by iteratively forming one vector pack at a time following a particular instruction traversal policy:

- **Bottom-up Traversal.** The learnt policy starts making packing decisions traversing the function in reverse, starting from the final instruction with valid packing candidates. At the end of each iteration $i$, we choose a predecessor of $I^i$ with valid packing candidates to consider for packing next.
- **Top-down Traversal.** The learnt policy starts making packing decisions traversing downwards from the first instruction with valid packing candidates. At the end of each iteration $i$, we choose a successor of $I^i$ with valid packing candidates to consider for packing next.

The instruction traversal policy selects a specific instruction $I^i$ in each iteration to be considered for packing next. The learnt vectorization policy decides which valid packing candidate instruction $I^j$ should $I^i$ be packed with. We define the components of our MDP as follows:

**State ($S_i$),** A state $S$ in our MDP is a tuple, $S = (I^i, PB)$. Here, $I^i$ represents the instruction selected by the fixed traversal order; $PB = \{P_1 \ldots P_k\}$ represents a set of formed packs.

**Start State ($S_1$),** $S_1 = (I^1, \{\})$. The current instruction to consider for packing is $I^1$ and there are no vector packs formed yet. The traversal policy decides $I^1$.

**Action ($A$),** On a given state $S = (I^i, PB)$, the set of legal [2] actions $A[S]$ is given as follows:
$$A[(I^i, PB)] = \{I^j \text{ such that } P_i = \{I^i, I^j\}, C_{\mathrm{I}}(P_i) \text{ and } \forall P_j \in PB, \, C_{\mathrm{II}}(P_i, P_j)\}$$

**Transition ($T$),** On a given state $(I^i, PB)$ and action $A = I$, the transition function $T$ simply adds the newly formed pack to $PB$: $T[(I^i, PB), I] = (I^{i+1}, PB \cup \{I^i, I\})$. The traversal policy decides on $I^{i+1}$. If there are no instructions with valid packing opportunities $I^{i+1} = I_\epsilon$.

**Reward ($R$),** We define the reward function as follows:
$$R((I^i, PB)) = \begin{cases} 0 & \text{if } I^i \neq I_\epsilon \\ F(PB) & \text{if } I^i = I_\epsilon \end{cases}$$

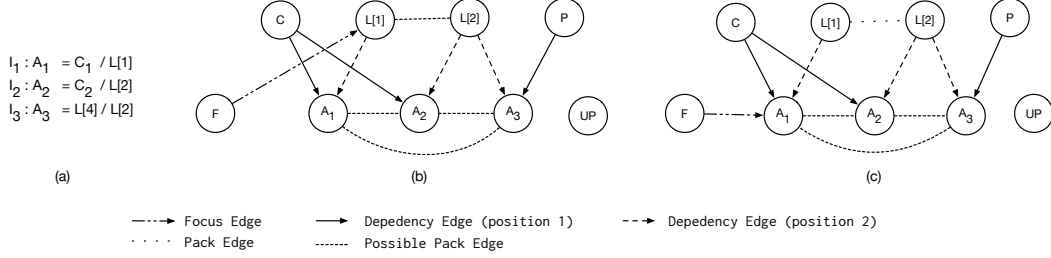

Figure 3: Graph formulation of MDP states for the code snippet shown in (a) under the forward traversal policy. Nodes: C-Constant, F-Focus, P-Pack, UP-Unpack and all other nodes are instruction nodes. Edges: different types are shown in the figure. Edges with arrows are directed. Figure (a) shows the initial state and Figure (b) shows the state after packing instruction nodes $\{L[1], L[2]\}$.

## 3.2 Graph Formulation of the MDP State

We use a Gated Graph Neural Network (GGNN) (Li et al., 2015) as part of the policy network modeling to make packing decisions for each state. To leverage the GGNN, we formulate the state of our MDP as a graph as follows:

**Nodes.** We consider 5 types of nodes to encode the graph features of a state $S_i$:
- **Instruction Node:** correspond to each instruction with at least one valid packing opportunity or instructions which are already packed.
- **Pack Node:** common node representing overhead packing instructions.
- **Unpack Node:** common node representing overhead unpacking instructions.
- **Constant Node:** common node representing any constant value used by instructions.
- **Focus Node:** special node that is connected to the instruction that is considered for packing in this iteration ($I^i$).

**Edges.** Following are the 4 types of edges connecting the above nodes:
- **Dependency Edge:** encodes if an instruction must be executed after another one in sequential order. More over, depending on the position of the arguments the instruction depends on, a different dependency edge type is created for a maximal of 5, with further additional arguments collapsed to the same dependency edge type 6. If a vector pack is formed and it requires overhead packing instructions a suitable dependency edge is added between the pack node and the two instruction nodes which form the vector pack. Similarly, if a vector pack needs to be used by a scalar a suitable dependency edge is added between the relevant instruction node and the unpack node. Note that, all dependency edges are directed.
- **Possible Pack Edge:** encodes whether two instructions can be packed together.
- **Packed Edge:** encodes instructions that are already packed together.
- **Focus Edge:** the focus edge connects the focus node to the instruction node that is considered for packing. This marks the node that we are making decisions on.

We illustrate our graph formulation for the code snippet shown in Figure 3(a) assuming a forward traversal policy. Figure 3(b) shows the initial state with focus edge directed at $L[1]$. Further, it shows edges for dependencies for each operand position, possible packs ($\{A_1, A_2\}$, $\{A_2, A_3\}$, $\{A_1, A_3\}$ and $\{L[1], L[2]\}$) as well as mandatory packing of non-adjacent load $L[4]$ which is used by $A_3$. Figure 3(c) shows the next state assuming pack $\{L[1], L[2]\}$ is formed. Notice the packed edge between $L[1]$ and $L[2]$ and the update of the focus edge to $A_1$, which is considered for packing next.

## 3.3 Neural Network Architecture

We use a GGNN to encode the aforementioned graph representation of the MDP state. GGNNs have shown promise in the field of code modeling (Brockschmidt et al., 2018), being able to capture intricate graph structures present in code sequences. GGNN maintains a hidden state vector for each node. During forward simulation, it passes some fixed round of messages between nodes through their connecting edges. The hidden state of a node is updated on each message passing iteration by first aggregating the messages passed from adjacent nodes and next by running the aggregate through a Gated Recurrent Unit (Cho et al., 2014) cell. Once message passing completes, we pass the hidden states of neighbors connected through possible pack edges of the node selected for packing through a multi-layer perceptron. Finally, we feed its output through a softmax layer to produce the action probabilities indicating how likely the selected node will be packed with its neighbors.

### 3.4 Imitation Learning

Vemal uses both supervised pre-training and imitation learning using the DAGGER algorithm (Ross et al., 2011) to imitate the packing decisions made by the ILP solver.

**Supervised Pre-training.** We first precompute optimal packing decisions using the ILP solver for the functions in our training set. We then pick a particular instruction traversal policy and create optimal trajectories of state-action pairs using the packing decisions made by the ILP solver, whilst updating the focus edge according to the traversal policy for each function. Here, the action is the selection of a node for packing among all valid packing candidates. The MDP state serves as the input to the GGNN based policy. We use cross entropy loss to train the policy by encoding the target action as a 1-hot vector. We perform supervised pre-training for a designated amount of batches.

**Imitation learning with DAGGER.** In the imitation learning phase, we sample a specific roll-out of the GGNN policy using the chosen instruction traversal policy for a randomly sampled batch of functions. For every MDP state visited by the GGNN policy, we encode the packing decisions already made into the ILP formulation, and use the ILP solver to provide optimal packings for the remaining instructions. We use this to find the optimal action for every state visited by the GGNN policy. The DAGGER algorithm augments the current dataset to include these state-action pairs. We then train on this augmented dataset similar to supervised pre-training. At each epoch, we continue to augment the dataset using the aforementioned strategy. This allows the GGNN policy to learn how to rectify its policy in cases where it falls out of the optimal trajectory.

### 3.5 Function Partitioning

Graph neural networks suffer from over-smoothing and scaling problems specially when the graph's diameter is large (Zhou et al., 2018). Real world functions can be arbitrarily sized, in some cases reaching more than 190000 instructions in our dataset. To alleviate scaling problems in our formulation, we partition functions based on their instruction counts and learn vectorization for each partition separately considering them as full functions using the formulation mentioned in Sections 3.1 to 3.4. Note that, we solve ILP problems considering each partition individually and use their solutions to train our agent. During inference time, we take separate rollouts using our agent for each partition of the function and finally merge all the decisions to form the final vectorization scheme.

## 4 Dataset

We use the same set of benchmark programs from goSLP (Mendis & Amarasinghe, 2018) to train and evaluate our imitation learning approach. Our dataset is composed of all individual functions collected out of the benchmark programs listed in Table 1. The benchmark programs represent floating-point C/C++ programs from SPEC2006 (Henning, 2006), SPEC2017 (Bucek et al., 2018) and NAS (Bailey et al., 1991) benchmark suites, which are well known benchmark suites used for evaluating compiler optimizations.

| Benchmark Suite | Benchmark Programs |
| --- | --- |
| SPEC2006 | 433.milc, 444.namd, 447.dealII, 450.soplex, 453.povray, 470.lbm, 482.sphinx3 |
| SPEC2017 | 508.namd_r, 510.parest_r, 511.povray_r, 519.lbm_r, 538.imagick_r, 544.nab_r |
| NAS | BT, SP, LU, MG, FT, CG, EP |

Table 1: Benchmark programs used for training and testing our learnt agent

### 4.1 Collection

We first compiled each source file to LLVM's intermediate representation (IR) just before LLVM's existing SLP vectorizer runs. By this way, we obtain the same IR that would have been seen by the vectorizer during an end-to-end compilation. Each source file has a number of functions and goSLP builds ILP problems considering a single function as the vectorization unit. Hence, we collected both the compiled LLVM IR (just before SLP vectorization) as well as the corresponding pairwise packing opportunities for each function for all programs in our benchmark suite.

### 4.2 Preparation

Using the methodology outlined in Section 4.1, we collected 35635 functions in total. However, only 3981 (11.17%) functions are vectorized by goSLP. If we use all collected functions during training, it induces a natural bias towards not vectorizing due to the imbalance in our dataset, even for functions

with abundant vectorizable opportunities. The goal of our learnt agent is to mimic goSLP as closely as possible when there are vectorizable opportunities. In cases where our learnt agent suggests an unprofitable scheme, it can be eliminated by a cost model similar to the current LLVM SLP vectorizer.

This asymmetric learning objective and the imbalance in our collected functions motivated us to create the final dataset that is biased towards functions which are vectorized by goSLP. We select all functions which are vectorized by goSLP as well as a random subset of non-vectorized functions such that, 80% (3981) of our dataset has functions with profitable vectorization schemes and 20% (995) do not. Finally, we split the dataset into a training set (80%) and a test set (20%) such that the proportionality of the vectorized and non-vectorized functions remains the same for both. There are 3169 and 812 vectorized functions in our training and test respectively. Our training set does not include any functions from the NAS benchmark suite which we use for evaluating the end-to-end runtimes of our learnt policy.

We evaluate 2 partitioning sizes, namely partitioning each function at 100 and 200 instruction counts. For each such partition, we create a new function with only the instructions from that partition and solve an ILP problem (goSLP's formulation) to retrieve the set of optimal actions for the partition. We report the final training and test set compositions for each partitioning scheme in Table 2.

| Scheme Name | Partition Size | Train set (partitioned functions) | | Test set (partitioned functions) | |
|---|---|---|---|---|---|
| | | vectorized | non-vectorized | vectorized | non-vectorized |
| p100 | 100 | 7203 | 1802 | 1776 | 446 |
| p200 | 200 | 5378 | 1346 | 1357 | 341 |

Table 2: Partitioned dataset statistics

# 5 Training and Testing

We now explain how we train Vemal's GGNN policy (Section 5.1) and use it in inference (Section 5.2) for making final vectorization decisions.

## 5.1 Training Setup

We learn the GGNN policy using the training set for each partition size for both backward and forward instruction traversal policies. Initially, there are 144944 and 163618 optimal state-action pairs for partition sizes 100 and 200 respectively under forward traversal policy. Under backward traversal the respective numbers are 144026 and 163448.

We pre-train each network using 3000 randomly sampled batches. At the beginning of each epoch, we randomly sample 400 of partitioned functions and augment our dataset using rollouts obtained for those functions. We use a mixed student-teacher policy similar to that used by the original DAGGER algorithm (Ross et al., 2011) to take rollouts, with the probability of choosing the teacher agent (ILP solver) exponentially decayed by $\beta = 0.9$ at the beginning of each epoch. Finally, we use goSLP's ILP solution to compute the optimal actions for each state we encounter during rollouts.

We use 20 message passing iterations in our GGNN. We train the neural network using stochastic gradient descent with momentum of 0.9, initial learning rate of 0.002 and with an exponentially decaying learning rate schedule (decay of 0.95). We randomly sample 50 state-action pairs for each batch and sample $\frac{\text{replay buffer}}{\text{batch size}}$ number of batches for each epoch.

## 5.2 Evaluation Criteria

In order to evaluate whether our trained policy is better than LLVM's SLP vectorization algorithm, we use three different metrics in our experiments.

- Average cost reduction across all vectorized functions in the test set compared to scalar.
- Geometric mean speedup across all vectorized functions in the test set compared to scalar.
- Geometric mean speedup of actual runtimes for NAS benchmark suite over LLVM SLP.

For the first two metrics we use values reported by LLVM's cost model. For the final metric we use actual wall clock times. For each partition size and instruction traversal policy, we evaluate each policy both when it uses the action with the highest probability (argmax policy) for each state as well as when it uses the best trace among $n$-rollouts (multi-rollout policy).

# 6 Experimental Results

We trained all four agents – partition sizes 100 and 200 and instruction traversal policies forward and backward – for 40 epochs. We use LLVM SLP (clang–6.0), goSLP and a random packing agent as our baselines for comparison. Note that, we restrict goSLP and our learnt agents to only perform pairwise packing (vectorization factor = 2), where as for LLVM SLP we do not restrict its vectorization factor and use its implementation without any change. Random packing agent is an agent which chooses uniformly from alternative actions for a given MDP state.

## 6.1 Static Results

Table 3 shows the average cost reduction and geometric mean speedup for the functions in the test set which are vectorized by goSLP (812). We include two additional comparison points, goSLP-p100 and goSLP-p200, which are policies that solve multiple ILP problems for partitioned functions using goSLP's formulation and merges the decisions to perform final vectorization.

| Vectorization Policy | Traversal Policy | # of rollouts | Average Cost Reduction (LLVM cost model) | Geo-mean Speedup (LLVM cost model) |
|---|---|---|---|---|
| goSLP | | | 23.6182 | 1.1226 |
| goSLP-p100 | | | 19.4815 | 1.1148 |
| goSLP-p200 | | | 21.2857 | 1.1193 |
| LLVM SLP | | | 12.1872 | 1.0873 |
| random | forward | | 1.0320 | 1.0150 |
| random | backward | | 1.0567 | 1.0126 |
| p100 | forward | 1 | 13.3633 | 1.0833 |
| p100 | forward | 10 | **14.9421** | 1.1018 |
| p100 | backward | 1 | 9.2180 | 1.0685 |
| p100 | backward | 10 | 11.3227 | 1.0911 |
| p200 | forward | 1 | 13.5259 | 1.0829 |
| p200 | forward | 10 | 14.7340 | **1.1020** |
| p200 | backward | 1 | 9.5801 | 1.0693 |
| p200 | backward | 10 | 11.5631 | 1.0912 |

Table 3: Average cost reduction and geometric mean speedups for vectorized functions in our test set based on LLVM's cost model under different vectorization policies

We notice that the agents using forward traversal learn a better vectorization policy than when using backward traversal. Also, all learnt agents except agents using backward traversal surpass LLVM SLP's average cost reduction. This fact is magnified with more rollouts, showing the efficacy of our learnt policy compared to LLVM SLP's greedy algorithm. The best performing agent (p100 with 10 rollouts) has an average cost reduction compared to scalar which is 22.6% higher than that of LLVM. This is in spite of the fact, LLVM is not restricted to pairwise packing.

Also, notice that partitioned versions of goSLP achieve a lower average cost reduction compared to that of goSLP. This is because of the sub-optimality introduced by solving subproblems as opposed to solving vectorization for the entire function. The maximum average cost reduction of learnt agents p100 and p200 are capped at those of goSLP-p100 and goSLP-p200 respectively. We should therefore, expect p200 to learn a better policy than p100. However, the learnt policies only have small overall average cost reduction differences. This is because the GGNN is not as good at approximating goSLP's packing policy when it comes to larger graphs. This gives rise to a trade-off space between sub-optimality of the solution and the learnability of a packing strategy at various partition sizes.

## 6.2 Runtime Results

We use our learnt GGNN policy for both partition sizes 100 and 200 to perform end-to-end vectorization for the NAS benchmark suite. We use the agents learnt under forward traversal and use both argmax and multi-rollout policy to evaluate the efficacy of the learnt policy on end-to-end runtimes. All benchmark programs are run on a Haswell Intel(R) Xeon(R) CPU E5-2680 v3 machine running at 2.50GHz with 32kB L1 and 256kB L2 cache sizes. We use Class-A workloads in our evaluation.

We run each benchmark program 3 times and report the median as is the common reporting method for compiler benchmarks. Figure 4 shows the runtime speedups for each program under goSLP and our learnt policies compared to LLVM SLP. The table shows the final geometric mean speedup for all policies compared to LLVM SLP.

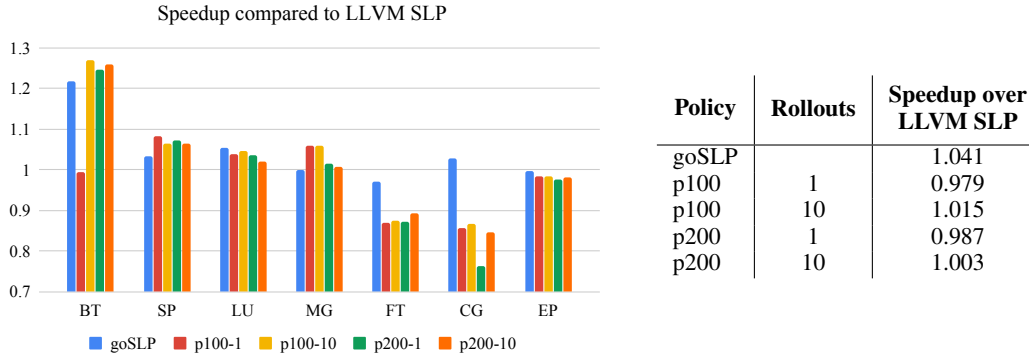

Figure 4: Speedup of goSLP, p100 with 1 and 10 rollouts, p200 with 1 and 10 rollouts compared to LLVM SLP for individual benchmarks in the NAS benchmark suite. The table shows the geometric mean speedups for the entire benchmark suite.

The best performing learnt agent achieves $1.015\times$ geometric mean speedup over all NAS benchmarks. In fact, both p100 and p200 learnt agents were able to beat LLVM SLP's performance with only 10 rollouts. This shows the efficacy of our learnt agents on end-to-end runtime performance.

More notably, all agents except p100 with 1 rollout beat or replicate the substantial runtime speedup of goSLP on BT benchmark over LLVM SLP. This signifies that the agents have learned a non-trivial vectorization policy not covered by LLVM SLP. Also, note that for SP and MG benchmarks all agents consistently beat goSLP in terms of performance. Even though goSLP performs optimal packing according to LLVM's cost model, for these benchmarks there exist other vectorization strategies, as uncovered by our learnt agents, which are better in terms of runtimes, but are suboptimal according to LLVM's cost model. This shows inaccuracies of the LLVM cost model. This provides evidence to our hypothesis, that in future a reinforcement learning based policy with a better cost model has the potential to learn an even better end-to-end vectorization policy than goSLP.

# 7 Related Work

Since the inception of vector machines, loop vectorization (Allen & Kennedy, 1987; Nuzman & Zaks, 2008) has been introduced to speedup scientific computing kernels. Larsen & Amarasinghe (2000) introduced superword level parallelism based vectorization targeting shorter SIMD width machines. Subsequently many techniques have emerged suggesting better heuristics to perform SLP vectorization (Porpodas & Jones, 2015; Porpodas et al., 2015; Liu et al., 2012; Shin et al., 2003, 2005, 2002). Recently, Mendis & Amarasinghe (2018) introduced a pairwise optimal packing algorithm using an ILP solver for statement packing which outperforms previous greedy approaches.

There has been work to find better compiler heuristics using machine learning (Stephenson et al., 2003; Cummins et al., 2017). More specifically, there has been previous attempts at identifying better heurisitics or program orders for vectorization using machine learning (Stock et al., 2012). However, it does not provide an end-to-end learnt solution for vectorization. Reinforcement Learning has been used to perform compiler instruction scheduling (McGovern et al., 2002) prior to the era of deep neural networks. In this work, we have shown how to imitate an ILP based solution using a graph neural network based policy to come up with the first end-to-end learnt auto-vectorizer.

In our formulation, we solve a NP-hard packing problem. Previously, reinforcement learning has been used to solve combinatorial optimization problems (Dai et al., 2017; Bello et al., 2016; Li et al., 2018). Comparatively, we have stronger supervision through an oracle with optimal actions.

# 8 Conclusion

Compiler auto-vectorization allows compilers to harness fine-grained parallelism within programs. Many greedy heuristic based solutions were proposed, and recently Mendis & Amarasinghe (2018) introduced a tractable solution with optimality guarantees using an ILP solver. Our work shows the feasibility of learning an end-to-end vectorization policy by imitating this optimal solution and we show that it outperforms well-tuned compiler heuristics used by the LLVM compiler. This holds out the promise that learnt compiler optimizations can be a better alternative to hand-written counterparts in the near future.

## Acknowledgements

We would like to thank Darsh Shah who was initially involved with this project and all reviewers for insightful comments and suggestions. This research was supported by DARPA D3M Award #FA8750-17-2-0126, DARPA HACCS Award #HR0011-18-C-0059 and DARPA SDH Award #HR0011-18-3-0007. Any opinions, findings, and conclusions or recommendations expressed in this material are those of the authors and do not necessarily reflect the views of the funding agencies.

## Footnotes

[1]https://software.intel.com/en-us/articles/introduction-to-intel-advanced-vector-extensions

[2] satisfies $C_{\mathrm{I}}$ and $C_{\mathrm{II}}$

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
