[Reviews · NeurIPS 2019]

Reviewer 1



This paper presents an auto-vectorization technique based on imitation learning. The authors claim that the proposed method is much faster than ILP-based method with some small loss in vectorization performance. The technique seems reasonable and the experiments do show its advantages over other techniques. However, I feel the paper is unsuitable for Neurips for the following reasons: 1) Auto-vectorization is an important problem in the compiler area. The paper should be more suitable to compiler/parallelization conferences such as PACT or CGO. 2) The idea of using imitation learning to make approximate decisions is not new. 3) The experiments are superficial. There is no comparison of the actual compile time and execution time with existing methods.

Reviewer 2



** I have read the author response and my opinion remains the same.** This paper uses imitation learning to solve the compiler auto-vectorization problem. It trains an agent to mimic the optimal solution generated by an integer linear programming solver. It outperforms production-level compiler LLVM in the experiments. Originality: The novelty is incremental. This paper directly combines well-known techniques and does not make any new contribution from the machine learning perspective. Quality: The experiment results look promising but it lacks detailed explanation. Only two figures were provided. Some case studies of the learned policy and more detailed results (more tables and plots) are expected. The claim "The learned policy runs faster then goSLP's ILP solver" is not backed by any experiment results. The author needs to provide a wall-clock time cost comparison of different methods. Clarity: The paper is clearly written and well organized. All required background is provided. The only drawback is that there is no mathematical description of the used algorithms such as GGNN and DAGGER. The author should provide these descriptions to make the paper accurate and clear. Significance: This paper shows a successful application of imitation learning for compiler optimization. It is a good trend for the compiler research community to use more intelligent data-driven machine learning algorithms. Overall this is an okay paper with limited novelty. The evaluation part definitely needs more experiments and result plots.

Reviewer 3



The authors propose to use graph neural networks to learn to imitate an optimal auto-vectorizer (ILP). This work is not the first to apply machine learning to auto-vectorization, however, the proposed approach is an end-to-end model for vectorization, whereas prior work focused on predicting performance. While the proposed approach does not match the optimal solution, it is able to outperform the heuristics of LLVM in a polynomial runtime. The authors run experiments comparing different imitation learning algorithms to ILP and LLVM. In the imitation learning experiments, different weights are placed on teacher vs student rollouts, and different node traversal strategies are proposed (forward vs backward), and benchmark their results on 3 datasets.

[Author Response · NeurIPS 2019]

We thank all reviewers for their valuable comments. We address the concerns raised by them below.

**Common Concerns**

**Novelty**

**Reviewer-2** *The novelty is incremental. This paper directly combines well-known techniques and does not make any*
*new contribution from the machine learning perspective.* The main contributions of our work is to show how to reduce
compiler auto-vectorization to a sequential decision making problem and how to solve it by learning a gated graph
neural network (GGNN) based policy using imitation learning. Based on our knowledge, this is the first time a compiler
pass is learnt end-to-end from data. Further, in our evaluation, we discuss how different traversal orders affect the
efficacy of the learnt policy, which gives insights to designing imitation or reinforcement learning algorithms in the
compiler / program transformation domain. Hence, we consider that our work is a valuable addition to NeurIPS under
the applications track.
**Reviewer-1** *The idea of using imitation learning to make approximate decisions is not new.* Imitation learning has been
successfully demonstrated to work in autonomous driving etc. Based on our knowledge, there has been limited work
on successfully applying it to discrete decision making problems. In fact, optimal subset selection problem reduces
to compiler auto-vectorization (Section 1), which is known to be NP-hard. We are using imitation learning to learn a
policy to approximately solve this NP-hard problem.

**Experimental Evaluation**

**Reviewer-1** *The experiments are superficial. There is no comparison of the actual compile time and execution time with*
*existing methods.* **Reviewer-2** *.... The author needs to provide a wall-clock time cost comparison of different methods.*
The main purpose of our work is to show that an end-to-end learnt policy guided by optimal decisions can outperform
traditional compiler heuristics. We evaluate the policy on real world programs (Figure 3 and 4) and show that we are
outperforming LLVM handily in terms of its own cost model. We expect this to translate to actual execution times,
since the cost model was designed to reflect how fast code would run on real hardware. Integrating the learnt policy
inside the compiler does not involve further research, but however is a considerable engineering challenge, which we
are currently undertaking and is necessary to get end-to-end execution times. We will include them in the final version.
Nonetheless, our results clearly show that we learn a better policy than the heuristics used by LLVM.

**Case Studies**

**Reviewer-2** *The author should provide more detailed experiment results, such as case studies of some packing strategies.*
**Reviewer-3** *What are corner cases that LLVM does not optimize, but this solution does?* One interesting case where
the learnt policy outperforms LLVM's heuristics is that it prefers to vectorize floating point divisions even with high
packing and unpacking overhead. When we individually time the functions, we find indeed this is beneficial, since
floating point division is an expensive operation and doing it in parallel reduces execution time. The learnt policy
picked up this fact entirely from data. Further, there are other more involved patterns which the learnt policy prefers to
vectorize which we will include in the final version.

**Reviewer Specific Questions**

**Reviewer-1** *Auto-vectorization is an important problem in the compiler area. The paper should be more suitable*
*to compiler/parallelization conferences such as PACT or CGO.* We came up with a novel GGNN formulation that
allows us to learn a policy to select profitable instruction packing opportunities. Additionally, we discuss how different
traversal orders of the GGNN formulation affects the overall policy. This gives insights on how to design imitation or
reinforcement learning algorithms in the compiler / program transformation domain. Taking all this into account, we
consider our work a self-contained solution that is worthy of publishing at NeurIPS in the applications track.

**Reviewer-2** *... no mathematical description of the used algorithms such as GGNN and DAGGER...*
We will include the mathematical formulations for GGNN and the DAGGER algorithm in the final version of the paper.
*The author should show how transferable the learned agent is.* The learnt model is transferable to programs in the
hidden test set, which consists of well-known benchmark programs from the SPEC benchmark suites.

**Reviewer-3** *Extend to cases where vector operations are not necessarily preferential.... by learning from real data or*
*building a model of the hardware.* This is indeed an interesting future direction which we are planning to pursue.
*does depth of the neural network/e.g. message-passing steps matter?* We find that the residual connections in the
GGNN formulation is important to ensure we learn a generalizable policy. We will discuss about this more in the final
version. The depth of the neural network does not matter that much.
*why is the model inferior to the optimal solution?* We are imitating an optimal solution, therefore the model is upper
bounded by the optimal solution and hence it should be inferior. However, we are investigating the gap between the
optimal solution and the learnt policy and ways to reduce it.
*Would more data or a higher capacity model help?* We hypothesize more data would help learning and generalizability.

[Meta-Review · NeurIPS 2019]

The paper is interesting and promising but the three reviewers do agree that this is a borderline paper around the acceptance threshold. The main concerns they have expressed are the following: (i) "incremental" application of reinforcement/imitation learning, (ii) weak experimental work and unsufficient benchmarking to other methods. I personally find it has its merits as an end-to-end application of imitation learning in quite an original context and I think it should be confronted to other application papers in the same domain (if any). Now to answer the points made by the reviewers, I think that (a) modeling the learning agent in this context is far from incremental as it combines intimate knowledge of compilers and solid mastering of graph theory and machine learning methodology; as a matter of fact, it certainly brings a paradigm shift in that community, (b) their experiments cover the case of instruction packing which was very recently related to ILP and they discuss the performance of their method wrt to optimal methods and to existing compilers, which is the best they can do at this stage. Given these two points, I tend to bump up slightly the assessment made by Reviewers #1 an #2 and push for acceptance.